# *Salvia miltiorrhiza* Bunge as a Potential Natural Compound against COVID-19

**DOI:** 10.3390/cells11081311

**Published:** 2022-04-12

**Authors:** Simon J. L. Petitjean, Marylène Lecocq, Camille Lelong, Robin Denis, Sylvie Defrère, Pierre-Antoine Mariage, David Alsteens, Charles Pilette

**Affiliations:** 1Louvain Institute of Biomolecular Science and Technology, Université Catholique de Louvain, 1348 Louvain-la-Neuve, Belgium; simon.petitjean@uclouvain.be (S.J.L.P.); david.alsteens@uclouvain.be (D.A.); 2Pole of Pneumology, ENT and Dermatology, Institute of Experimental and Clinical Research (IREC), Université Catholique de Louvain, 1348 Brussels, Belgium; marylene.lecocq@uclouvain.be; 3BOTALYS, 8 Avenue des Artisans, 7822 Ghislenghien, Belgium; c.lelong@botalys.com (C.L.); r.denis@botalys.com (R.D.); s.defrere@botalys.com (S.D.); pa.mariage@botalys.com (P.-A.M.); 4Walloon Excellence in Life Sciences and Biotechnology (WELBIO), 1300 Wavre, Belgium; 5Department of Pneumology, Cliniques Universitaires Saint-Luc, 1200 Brussels, Belgium

**Keywords:** SARS-CoV-2, COVID-19, ACE2, infection, viral entry, binding inhibitor, AFM, atomic force microscopy, force spectroscopy, single-molecule

## Abstract

*Salvia miltiorrhiza* Bunge, commonly called danshen, is widely used in traditional Chinese medicine for its cardiovascular and neuroprotective effects, which include antioxidative, anti-inflammatory, and antifibrotic properties. The purpose of this study was to evaluate the preclinical potential of *S. miltiorrhiza* extracts for the treatment of COVID-19. First, the impact of the extract on the binding between SARS-CoV-2 and the cellular ACE2 receptors was assessed using atomic force microscopy (AFM), showing a significant reduction in binding by the extract at concentrations in the µg/mL range. Second, the interference of this extract with the inflammatory response of blood mononuclear cells (PBMCs) was determined, demonstrating potent inhibitory properties in the same concentration range on pro-inflammatory cytokine release and interference with the activation of NFκB signaling. Together, these in vitro data demonstrate the potential of *S. miltiorrhiza* against COVID-19, consisting first of the blockade of the binding of SARS-CoV-2 to the ACE2 receptor and the mitigation of the inflammatory response from leukocytes by interfering with NFκB signaling. This dataset prompts the launch of a clinical trial to address in vivo the clinical benefits of this promising agent.

## 1. Introduction

According to the World Health Organization (WHO), COVID-19 (coronavirus disease 2019) is an infectious disease caused by the SARS-CoV-2 (severe acute respiratory coronavirus-2). This new virus and disease were unknown before the outbreak began in Wuhan in December 2019. COVID-19 has now become a pandemic disease, with over 479 million cases and 6 million deaths reported globally (WHO; weekly epidemiological update—29 March 2022).

So far, no drug has been shown to be really effective in treating COVID-19 during the acute phase, notably in disease caused by new variants escaping vaccine-induced immunity or in immunocompromised persons in whom vaccination is poorly effective. To this end, herbs used in traditional Chinese medicine (CM) present a potentially valuable resource [1]. Previously, in 2003 and in 2009, CM approaches were used to prevent or treat severe acute respiratory syndrome (SARS) and H1N1 influenza, respectively [2,3].

Among medicinal plants, *Salvia miltiorrhiza* Bunge, commonly called danshen, is attracting much interest as it is widely used in traditional CM for its cardiovascular and cerebrovascular effects, and its antioxidative, neuroprotective, antifibrotic, anti-inflammatory, and antineoplastic activities [4]. Lipophilic diterpenoids (mainly tanshinones) and hydrophilic phenolic compounds (such as salvianolic acids) are the major active compounds found in its roots [4,5,6]. Tanshinones are a group of abietane diterpenes which includes dihydrotanshinone I (DHt), tanshinone I (TI), cryptotanshinone (CTt), and tanshinone II A (TII). They all share relevant biological activities, such as antioxidant, anti-inflammatory, antibacterial, and even antitumoral properties [5].

Several studies have pointed out that *S. miltiorrhiza* could have a therapeutic potential for COVID-19. The first study performed a screening of CM known to be biologically active against SARS coronavirus or Middle East respiratory syndrome coronavirus and performed a docking analysis to evaluate their bonding potential against three SARS-CoV-2 proteins (i.e., the spike (S) protein, the 3C-like protease (3CLpro), and the Papain-like protease (PLpro)). These three proteins are essential in the virus cycle during viral entry or replication, make them attractive targets for drug development. Of the 13 natural compounds with potential SARS-CoV-2 activity, three active compounds of *S. miltiorrhiza*, (CTt, DHt, and TII) were found [7]. In addition, tanshinones derived from *S. miltiorrhiza* were already found to be specific and selective inhibitors for 3CLpro and PLpro [8]. Furthermore, three salvianolic acids (salvianolic acid A (SAA), salvianolic acid B (SAB), and salvianolic acid C (SAC)) can inhibit the entry of SARS-CoV-2 into cells by binding to the receptor-binding domain (RBD) of the SARS-CoV-2 spike protein and the specific host cellular receptor enabling its entry—angiotensin-converting enzyme 2 (ACE2) [9]. Finally, *S. miltiorrhiza* may also have effects that could be relevant for advanced or late manifestations of the disease, such as inhibition of epithelial-to-mesenchymal transition as well as attenuation of bleomycin-induced experimental pulmonary fibrosis by CTt [10] and by TII, through modulation of the ACE2/angiotensin-(1–7) axis [11].

To evaluate the preclinical potential of *S. miltiorrhiza* extracts for the treatment of COVID-19, the effect of this extract on the binding between SARS-CoV-2 and the cellular ACE2 receptors was monitored using atomic force microscopy (AFM), showing a significant reduction in the binding at concentrations in the µg/mL range. Second, the interference of the extract with the inflammatory response of blood mononuclear cells was assessed, also showing significant inhibitory effects of the extract in the same concentration range. Together, these in vitro data demonstrate that *S. miltiorrhiza* extracts display beneficial effects on both virus entry and regulation of the inflammatory response.

## 2. Material and Methods

### 2.1. The S. miltiorrhiza Root Extract

The *S. miltiorrhiza* roots were hydroponically cultivated using an innovative vertical farming technology, with a strict control of growing conditions, by BOTALYS (Ghislenghien, Belgium). The extract was prepared from roots according to a proprietary process. Briefly, the roots were mechanically extracted following a phase extraction (with saline buffer; pH 5.5–6.0). The tanshinones-enriched extract (lipophilic fraction) was then recovered, dried, and stored at −20 °C.

For the present in vitro experiments, the extract was resuspended into dimethylsulfoxide (DMSO, Merck) at the desired concentration.

### 2.2. Analysis of the S. miltiorrhiza Root Extract (UHPLC)

Compound extraction was performed as follows: The dry extract (50 mg) was extracted three times with 70% methanol (2 mL; 1.5 mL; 1.5 mL) under ultrasonication for 15 min and centrifugation at 14,000× *g* for 5 min. The supernatant collected was filtered through a 0.45-mm Millipore filter and used for ultra-high performance liquid chromatography (UHPLC) analysis.

UHPLC analysis was conducted using a SHIMADZU UHPLC LC20 ADXR, which is a modular system consisting of Detector SPD-40V, Autosampler SIL-40C, Pump LC-40B XR, and Column oven CTO-40C with column Shim-pack GIST C18 2 µm (150 × 2.1 mm). The mobile phase was made up of water + 0.1% phosphoric acid and acetonitrile (solvent B). Each run started at 20% B, increasing to 35% B at 10 min, 40% B at 20 min, 70% B at 65 min, decreasing to 20% B at 80 min, 5% B at 85 min, increasing to 20% at 90 min, then keeping this composition steady until finishing. The column temperature was set to 25°C and flow rate to 0.20 mL/min. Injection volume was 10 µL UV detection, set to a wavelength range of 190 to 800 nm. Standards of metabolite compounds were purchased (rosmarinic acid and salvianic acid or danshensu from Sigma Aldrich (Saint-Louis, MO, USA); ursolic acid, salvianolic acid B, Tanshinone IIa, Tanshinone IIb, Tanshinone VI, dihydrotanshinone and Tanshinone I from Carbosynth.; Tetrahydrotanshinone from Chemfaces) and calibration curves were performed.

### 2.3. Preparation of ACE2-Coated Model Surfaces

The ACE2 molecule (SinoBiological, Beijing, China; 90211-C02H) was grafted onto gold-coated model surfaces taking advantage of the NHS-EDC chemistry. The gold substrates were first washed with ethanol, dried with gas nitrogen, and cleaned by UV-O treatment (Jetlight, Irvine, CA, USA) for 15′. Then, they were incubated overnight in an alkanethiol solution (99% 11-mercapto-1-undecanol 1 mM (Sigma Aldrich) and 1% 16-mercaptohexadecanoic acid 1 mM (Sigma Aldrich) in ethanol). After rinsing with ethanol and drying with nitrogen gas, the samples were immersed in a solution of 25 mg/mL 1-ethyl-3-(3-dimethylaminopropyl)carbodiimide (Sigma Aldrich) and 10 mg/mL N-hydroxysuccinimide (Sigma Aldrich) in milliQ water for 30′. Finally, the samples were rinsed in milliQ water and a drop of the ACE2 protein (0.1 mg/mL in PBS) was pipetted on them. After a 1 h incubation, the ACE2-coated model surfaces were rinsed with PBS and ready for AFM measurements (performed the same day as functionalization).

### 2.4. Tip Functionalization

MSCT-D tips (Bruker) were functionalized with either the S1-subunit of the SARS-CoV-2 spike glycoprotein (Genscript, Piscataway, NJ, USA; Z03501) or UV-inactivated full virions (provided by Prof. S. Boulant and Dr. Megan Stanifer, University of Heidelberg, Germany).

The silicon nitride tips were first cleaned in chloroform (3 times 5′) and by UV-O treatment for 15′ (Jetlight). Meanwhile, 3.3 g of ethanolamine hydrochloride was dissolved in 6.6 mL of dimethylsulfoxide (DMSO) in which the cantilevers were immersed overnight. They were then rinsed with DMSO and ethanol (three times 1′ each) and incubated for 2 h in the PEG linker solution (3.3 mg of NHS-PEG_24_-Ph-aldehyde in 0.5 mL chloroform) with 30 µL of triethylamine. After rinsing with chloroform (3 times 5′), the tips were placed on parafilm (Bemis NA) in a star conformation (tips pointing to each other) and a drop (50 µL) of S1 protein (0.1 mg/mL) or UV-inactivated SARS-CoV-2 (10^8^ particles/mL) was pipetted on them with a freshly prepared solution of NaCNBH_3_ (~6% wt. vol^−1^ in 0.1 M NaOH_(aq)_) for 1 h. Finally, 5 µL of ethanolamine 1 M (pH 8) was added for 10 s, the tips rinsed 3 times in PBS and stored in PBS at 4 °C until the experiments (max 2 days after functionalization).

### 2.5. Preparation of S. miltiorrhiza Stock Solutions

Stock solutions were prepared by dissolving *S. miltiorrizha* root powder extract in DMSO. For experiments with the S1-subunit only, the stock solution was made at 1 mg/mL and then dissolved in PBS to 1, 10, 50, and 100 µg/mL, respectively (leading to a final concentration of 0.1 to 10% in DMSO). For measurements featuring the full virion, the stock was made more concentrated (10 mg/mL) in order to reduce the final concentration in DMSO and thereby preserving protein integrity. The stock was dissolved to the same final concentrations of 1, 10, 50, and 100 µg/mL (associated with reduced DMSO concentrations between 0.01 and 1%).

### 2.6. FD Curve-Based Atomic Force Microscopy (AFM)

The assessment of the inhibition of the ACE2 binding was performed using an AFM Multimode 8 (Bruker, Santa Barbara, CA, USA; Nanoscope software v9.2) operating in the force volume (contact) mode. Interactions between functionalized MSCT-D probes (nominal spring constant of 0.03 N/m) and ACE2-coated model surfaces were probed on 5 × 5 µm areas, with a force setpoint of 500 pN, hold time of 250 ms, resolution of 32 × 32 pixels, and line frequency of 1 Hz.

Four consecutive maps were recorded for each experiment, with increasing extract concentration of 0, 1, 10, and 100 µM, respectively. Concretely, the first map was recorded in a PBS droplet that was then replaced by an extract solution of 1 µM, a new map recorded, and so on. Both tips and surfaces were changed for each experiment.

### 2.7. Isolation and Culture of Peripheral Blood Mononuclear Cells (PBMC)

Peripheral blood was collected from healthy donors with no sign of active infection or atopic disease. The study was approved by the Ethics Committee of the Cliniques Universitaires Saint-Luc (Brussels, Belgium), and all subjects gave written informed consent. PBMC were isolated after density gradient centrifugation (Lymphoprep^®^, Axis-Shield, Oslo, Norway) from heparin-anticoagulated blood samples. After centrifugation (800 g, 30 min), PBMC were isolated, washed, and resuspended in RPMI-1640 (Lonza, Basel, Switzerland) culture medium containing 10% heat-inactivated fetal bovine serum (Gibco Life Technologies, UK), 2 mM L-glutamine, 100 U/mL penicillin, and 100 µg/mL streptomycin (Lonza).

PBMC (0.5 × 10^6^ cells) were cultured in 24-well flat-bottomed plates in 0.5 mL of RPMI-1640 culture medium. Cells were incubated at 37 °C, 5% CO_2_ for 1 h with the *S. miltiorrhiza* extract (0, 0.5, 1, 5, or 10 µg/mL or DMSO control). They were then incubated with a medium or with R848 (0.02, 0.1, 0.5, 1, 2.5, or 5 μg/mL for the dose-response experiments and 1 µg/mL for the other experiments; Merck group) or with DMSO as control (Sigma, Saint-Louis, MO, USA) at 37 °C, 5% CO_2_ for 24 h, before collecting and freezing the supernatants (−20 °C) until cytokine assays.

### 2.8. Cytokine Release Assays

Interleukin (IL)-1β, IL-6, and TNF-α were measured by sandwich ELISA in the cell-culture supernatants using paired antibodies or Duoset kit (Biotechne R&D Systems, Minneapolis, MN, USA), IL-8/CXCL8 (Sigma–Biotechne R&D Systems, Minneapolis, MN, USA). IFN-α was also quantified by ELISA (PBL Assay Sciences, Piscataway, NJ, USA).

### 2.9. Western Blot for NF-κB Signalling

PBMC were stimulated by R848, with or without a pre-incubation step with the *S. miltiorrhiza* extract, as described above, in 24-well plates at 1 × 10^6^ cells per well. Cells were subsequently lysed for 30 min on ice in 150 µL of the RIPA lysis buffer (50 mM Tris-HCl, pH 7.4, 0.25% sodium deoxycholate, 150 mM NaCl, 1 mM EDTA, 1 mM PMSF, 1 mM Na3 VO4, 1 mM NaF). Lysates were clarified by centrifugation at 4 °C for 10 min at 10,000× *g.* Equal amounts of lysates were separated by 12% SDS-PAGE and electrotransferred onto a nitrocellulose membrane, subjected to immunoprobing using antibodies to phosphorylated p65NF-κB S536, acetylated p65NF-κB K310 (1:1000, rabbit monoclonal ab, Cell Signaling Technology, Danvers, MA, USA), total NF-κB p65 (1:1000, mouse monoclonal ab, Cell Signaling) or GAPDH (1:1000 rabbit polyclonal ab, Sigma, Saint-Louis, MO, USA). Secondary abs consisted of HRP-conjugated goat IgG anti-rabbit IgG (for p65NF-κB S536, acetylated p65NF-κB K310 and GAPDH; 1:2000, Cell Signaling Technology) or anti-mouse IgG (for total p65NF-κB 1:5000, Sigma). Immunoreactive bands were developed using chemiluminescence (Amersham^TM^ ECL, GE Healthcare, Buckinghamshire, UK) and detected with a Chemidoc XRS apparatus (Bio-Rad, Hercules, CA, USA). The intensity of each band was measured with the densitometry program Quantity One (Bio-Rad).

### 2.10. Statistics

Data were analyzed by means of the Kruskal–Wallis non-parametric test with Dunn’s correction for PBMC data using SPSS Statistics software V28.0.1. *p*-value for mean comparison was determined by unidirectional *t*-test in JMP and set at 0.05 for significance.

## 3. Results

### 3.1. Characterization of the S. miltiorrhiza Root Powder Extract

The content of the active compounds danshensu (or salvianic acid), rosmarinic acid, salvianolic acid A, salvianolic acid B, ursolic acid, tanshinone I, tanshinone IIA, tanshinone IIB, tanshinone VI, dihydrotanshinone I, cryptotanshinone, and tetrahydrotanshinone of the *S. miltiorrhiza* root powder extract were analyzed by UHPLC. The results are summarized in Table 1.

### 3.2. Assessment of the Inhibitory Effect of the Interaction between the S1-Subunit and ACE2

To evaluate the inhibitory effect of *S. miltiorrizha* extract on the SARS-CoV-2—ACE2 interaction, the interaction at the single-molecule level was monitored using AFM, as previously described and validated [12].

First, the anti-binding potential of the *S. miltiorrizha* extract was tested towards the interaction between the purified S1-subunit of the SARS-CoV-2 S protein and the ACE2 receptors. To this end, S1 was grafted on the AFM tip using an heterobifunctional linker and surfaces were coated with the purified ACE2 receptors (Figure 1A and Section 2). Single-molecule force spectroscopy (SMFS) experiments were performed by successive approach and retraction movement between the tip and the surface. From these cycles, force versus distance (FD) curves were recorded (Figure 1B). Upon retraction, specific unbinding events were observed corresponding to the rupture of the S1-ACE2 bond. The binding frequency is extracted and corresponds to the ratio between the number of FD curves showing specific adhesion event and the total number of FD curves recorded.

As the extract is solubilized in DMSO (see Section 2) that could potentially interfere with protein structure and stability, first the influence of DMSO was tested on the frequency of binding events. SMFS were performed at increasing concentrations of DMSO (matching the concentration of the injected extracts), revealing no significant decrease in the binding frequency in the range 0–10% of DMSO (Figure 1C).

Next, the extract was tested using increasing concentrations during successive AFM mappings (0, 1, 10, 50, and 100 µg/mL, respectively; N ≥ 3 independent experiments). The results show (Figure 1D) a significative drop in binding probability in the presence of *S. miltiorrizha* extract at a concentration of 1 µg/mL. A 50% inhibition was reached at concentrations ≥ 100 µg/mL, suggesting a IC50 in the same range.

### 3.3. Assessment of the Inhibitory Effect of the Interaction between the UV-Inactivated SARS-CoV-2 Virions and ACE2

In a second step, the influence of the plant extract was evaluated on the binding of UV-inactivated SARS-CoV-2 virions grafted onto the AFM tips (Figure 2A,B). Working directly on full virions enables to accommodate more physiological conditions, in particular the multivalency encountered during virus binding to cell surface receptors.

As seen previously, control experiments with increasing concentrations of DMSO revealed no significant effect on the binding frequency (Figure 2C). One has to note that more concentrated stock solutions of extract were used, allowing to reduce the final concentrations in DMSO (see Methods). In more physiologically-relevant conditions, the results demonstrate that *S.miltiorrhiza* show better inhibitory potential. Significant inhibition is also observed at a concentration of 1 µg/mL; however, the binding frequency is already reduced by 50% at a concentration ≈10 µg/mL.

### 3.4. Establishment of an In Vitro Cell-Based Assay of TLR7/8 Activation of PBMC Mimicking the Inflammatory Response to SARS-CoV Infection

In order to study the modulation of inflammatory responses, first an in vitro system was established by using activation by R848 (Resiquimod)—an agonist of TLR7 and TLR8—of PBMC from healthy blood donors. Incubation of PBMC with R848 induces the release of the pro-inflammatory cytokines IL-1β, TNF, IL-8/CXCL8, IL-6, as well as IFN-α in a dose-response manner, which reached significance for all cytokines (except IFN-α) at 1 µg/mL (Figure 3A–E). For IFN-α, a significant increase was only observed at 5 µg/mL due to a large variability in the response across donors, with a plateau reached at ≈ 0.5 μg/mL (Figure 3F).

### 3.5. Assessment of the Inhibitory Effect of a S. miltiorrhiza Extract on the PBMC Response to TLR7/8 Activation

In a second step, this in vitro system was used to assess whether *S. miltiorrizha* extract could regulate the response of PBMC to TLR7/8 activation. Following preincubation for 1 h of PBMC with the extract, a significant inhibition of IL-1β, TNF, and IL-6 was observed at a concentration of 5 or 10 µg/mL (Figure 4A–C), with a similar potency when compared to 1 μM dexamethasone, the current reference treatment for COVID-19. Hence, no significant effect was observed on IL-8/CXCL8 secretion (Figure 4D), whereas the release of IFN-α was completely suppressed at 5 μg/mL, achieving a stronger inhibition than 1 µM dexamethasone (Figure 4E).

### 3.6. Assessment of the Mechanism of Inhibition of the Cell Response to TLR7/8 Activation

The potential mechanisms by which the *S. miltiorrizha* extract could suppress cytokine responses was then investigated, focusing on the NFκB pathway as a master inflammatory signaling pathway. The phosphorylation of NFκB p65 (at serine 536) was observed at 30 min of incubation with R848, without alteration in its acetylated form (Figure 5A). Upon preincubation with the extract, inhibition of p65 phosphorylation was observed at 10 µg/mL (Figure 5B) with no effect on its acetylation (Figure 5C), indicating that *S. miltiorrizha* extract may interfere with NFκB signaling activation.

## 4. Discussion

The objective of this study was to investigate in vitro the therapeutic potential of *S. miltiorrhiza* extract in relation to COVID-19. First, using AFM to analyze at the single-molecule level the interaction between the S1 domain of the S protein and the cellular receptor ACE2, an anti-binding potential of the SARS-CoV-2–ACE2 interaction was shown by the extract at concentrations in the µg/mL range. This effect is even stronger in a more physiologically relevant context involving the full-length SARS-CoV-2 virion, illustrating the critical role played by multivalence in biological interactions. Second, the cellular TLR7/8 activation assay of PBMCs reveals that the extract has potent inhibitory properties on pro-inflammatory cytokine release and interferes with the activation of NFκB signaling.

Previous studies have already shown that the spike protein of the SARS-CoV-2 and the salvionic acids present in *S. miltiorrhiza* are able to bind to each other [9,13,14]. Similarly, a docking analysis demonstrated that dihydrotanshinone, another active compound of *S. miltiorrhiza*, can bind to the SARS-CoV-2 spike protein [7]. The present study proves that incubation with *S. miltiorrhiza* root extract results in a significant decrease in virion binding to its specific cellular receptor, ACE2. In turn, this loss of adhesion would undoubtedly lead to a decrease in SARS-CoV-2 infection, as the ACE2 receptor acts as a specific cellular entry receptor for the virus. This is in accordance with the study of Kim et al. who demonstrated that dihydrotanshinone was able to block MERS-CoV entry [15] and with the more recent study of Wang et al., who showed that Danshensu had a potent antiviral activity and a concentration-dependent manner in inhibiting SARS-CoV-2. Danshensu was also identified to potently inhibit the entry of SARS-CoV-2 into ACE2-overexpressed cells [16].

The second key feature of *S. miltiorrizha* identified in this study concerns its anti-inflammatory properties. For the first time, anti-inflammatory effects were highlighted on PBMC activated by R848 mimicking a viral infection, and revealed mechanical insights, namely that inhibition of the p65 phosphorylation is responsible of the interference with NFκB signaling activation. This anti-inflammatory action of *S. miltiorrizha* is consistent with previous studies of inflammatory diseases performed in vitro and in vivo in animal models [17]. Moreover, the present extract was specifically abundant in cryptotanshinone, dihydrotanshinone I, tanshinone IIA, and tanshinone IIB (see Table 1). The anti-inflammatory effects of these molecules were already evidenced in vitro and in vivo by inhibiting the NF-κB signaling pathway [18,19,20,21,22]. More specifically, in a model of pulmonary fibrosis, which is one of the major complications in COVID-19 patients, it was shown that compounds such as CTt and TII may mitigate bleomycin-induced pulmonary fibrosis caused by an excessive inflammatory reaction [10,11]. Furthermore, it was demonstrated that pretreatment with Danshensu alleviated pathological alterations in mice infected with SARS-CoV-2 (lung inflammation, cytokines in serum and lung, alteration of the antioxidant system), and this study highlighted that Danshensu effectively suppressed the activations of NF-κB p65 in the lung tissues of SARS-CoV-2-treated mice [16].

The in vitro nature of the experiments conducted represents a limitation of this study. This may lead to overestimation of the therapeutic potential, which might not be exerted in vivo to the same extent due to several reasons. More specifically, PBMC do not recapitulate completely the several actors of the inflammatory response in vivo. It has, however, now been accepted that blood monocytes represent a key pro-inflammatory cell during severe COVID-19 [23]. Altogether, despite those limitations, this study offers valuable preclinical information that could be used before designing clinical trials—to confirm and evaluate their therapeutic potential in human subjects.

## 5. Conclusions

This study demonstrates the potential of *S. miltiorrhiza* for COVID-19 treatment, consisting first of the blockade of the binding by the S1-subunit to the ACE2 receptor and of the mitigation of the inflammatory response from leukocytes by interfering with NFκB signaling activation. This dataset makes this extract a promising agent, acting potentially on both preventive or early phases of the infection and on the inflammatory response occurring later during its natural history. In addition, these in vitro data suggest that the extract may be as effective (and even more for IFN-α inhibition) than the current reference treatment of the disease, namely corticosteroids (in particular, dexamethasone). Furthermore, the strong inhibition of IFN-α appears to be a key point since this cytokine is critical in the pathophysiology of COVID-19 by upregulating ACE2 [24], strengthening the candidacy of *S. miltiorrhiza* as a potential natural compound against COVID-19.

## Figures and Tables

**Figure 1 cells-11-01311-f001:**
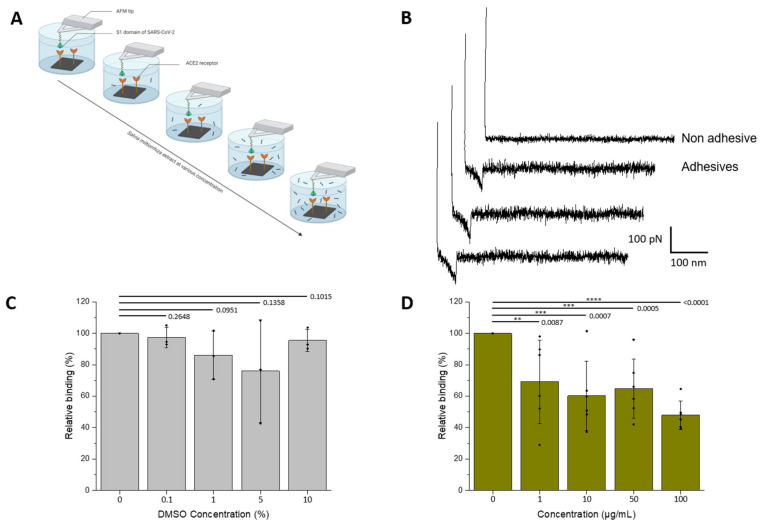
Assessment by AFM of the inhibitory effect of *S.miltiorrizha* extract on the interaction between the S1-subunit and ACE2. (**A**) The inhibitory potential of roots *S. miltiorrhiza* extract is assessed by measuring the binding probability between the S1 subunit immobilized on the AFM tip and the ACE2 receptor grafted on a model surface, before and after incubation with the extracts at increasing concentrations (0, 1, 10, 50, and 100 µM, respectively). (**B**) Examples of non-adhesive (above) and adhesives (below) recorded between S1-functionalized AFM tips and ACE2-coated model surfaces showing specific adhesion events. (**C**,**D**) Histograms showing the binding probability (BP, expressed in %) as a function of the *S. miltiorrhiza* extract or DMSO concentration. The error bar indicates s.d. of the mean value. One data point belongs to the BP from one map. The figure on panel a was created with Biorender.com.

**Figure 2 cells-11-01311-f002:**
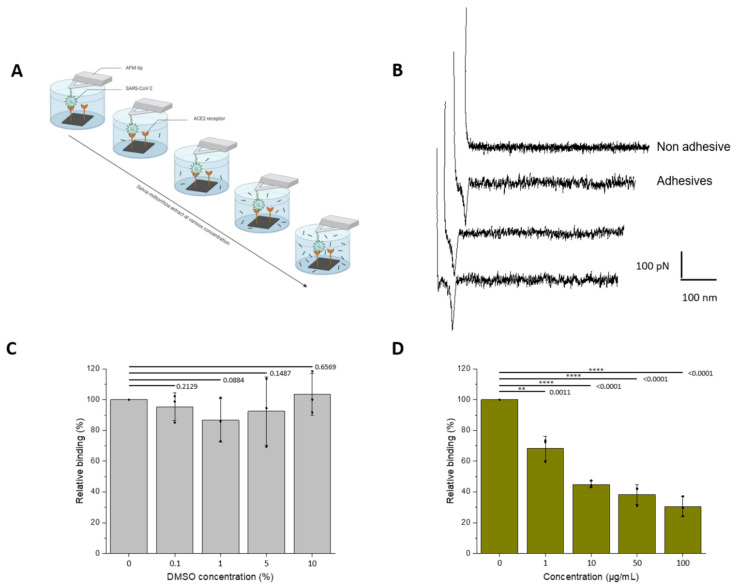
Assessment of the inhibitory effect of a *S. miltiorrizha* extract powder on the interaction between the UV-inactivated SARS-CoV-2 virions and ACE2 monitored by AFM. (**A**) The inhibitory potential of roots *S. miltiorrhiza* extract is assessed by measuring the binding probability between the UV-inactivated SARS-CoV-2 immobilized on the AFM tip and the ACE2 receptor grafted on a model surface, before and after incubation with the extracts at increasing concentrations (0, 1, 10, 50, and 100 µM, respectively). (**B**) Examples of non-adhesive (above) and adhesive (below) force distance curves recorded between UV-inactivated SARS-CoV-2 virions-functionalized AFM tips and ACE2-coated model surfaces showing specific adhesion events. (**C**,**D**) Histograms showing the binding probability (BP, expressed in %) with the different concentrations of extracts (0, 1, 10, 50, and 100 µM) or DMSO for the control (0, 0.01, 0.1, 0.5, and 1%). The error bar indicates s.d. of the mean value. One data point belongs to the BP from one map. The figure on panel a was created with Biorender.com (accessed on 10 March 2022).

**Figure 3 cells-11-01311-f003:**
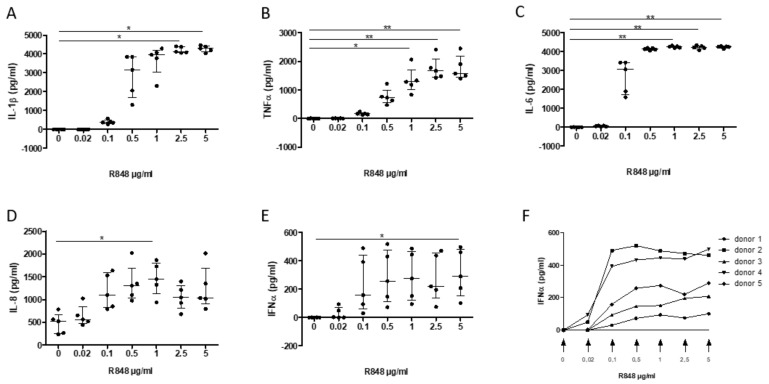
Dose-response release of interleukin (IL)-1β (**A**), TNF-α (**B**), IL-6 (**C**), IL-8 (**D**), and IFN-α (**E**–**F**) following PMBC activation by R848 (0.02, 0.1, 0.5, 1, 2.5, and 5 µg/mL; 0, DMSO control) on PMBC release of interleukin (IL)-1β (**A**), TNF-α (**B**), IL-6 (**C**), IL-8 (**D**), and IFN-α (**E**,**F**). Data are median ± interquartile range (*n* = 5 separate experiments; * *p* < 0.05, ** *p* < 0.01).

**Figure 4 cells-11-01311-f004:**
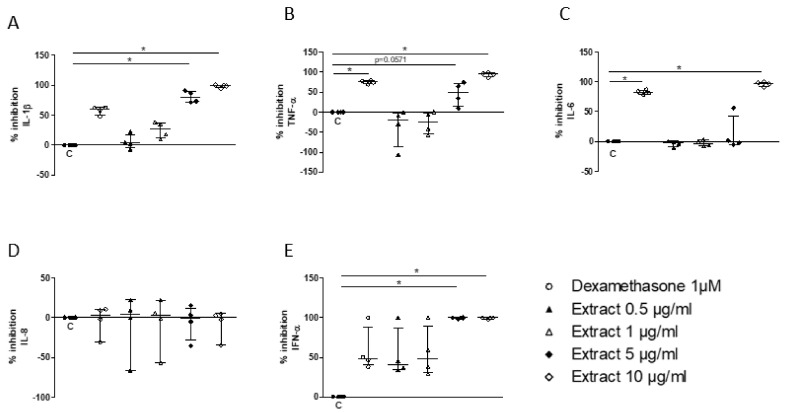
Inhibitory effect of *S. miltiorrhiza* extract (0.5, 1, 5, and 10µg/mL) on PBMC release of IL-1β (**A**), TNF-α (**B**), IL-6 (**C**), IL-8 (**D**), and IFN-α (**E**) upon stimulation by R848 (1µg/mL). Data are median ± interquartile range (*n* = 5; * *p* < 0.05).

**Figure 5 cells-11-01311-f005:**
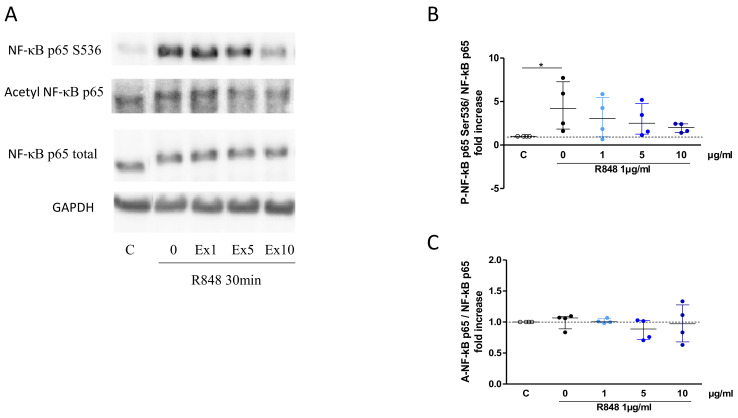
Inhibitory effect of *S. miltiorrhiza* extract (Ex; 1, 5 or 10 µg/mL) on NF-κB activation following incubation with R848 (1 µg/mL) for 30 min. Representative western blot (**A**), with quantification data for phosphorylation of p65NF-κB S536 (**B**) and acetylation of p65NF-κB K310 (**C**). GAPDH was used as loading control (*n* = 4; * *p* < 0.05).

**Table 1 cells-11-01311-t001:** Content of active compounds (expressed in mg/g of dry mater; except for the total expressed in %) measured in the *S. miltiorrhiza* root powder extract.

Active Compound	*S. miltiorrhiza* e Root Powder Extract (mg/g Dry Mater)
Danshensu/Salvianic acid	0.05
Rosmarinic acid	1.61
Salvianolic acid A	0.31
Salvianolic acid B	0.07
Ursolic acid	1.64
Tanshinone I	0.90
Tanshinone IIA	10.80
Tanshinone IIB	10.22
Tanshinone VI	5.53
Dihydrotanshinone I	15.35
Cryptotanshinone	77.59
Tetrahydrotanshinone	3.61
Total (%)	12.8%

## Data Availability

Not applicable.

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
