# Peer review of "Salvia miltiorrhiza Bunge as a Potential Natural Compound against COVID-19"

_cells, 2022, doi:10.3390/cells11081311_

Round 1
Reviewer 1 Report
The purpose of the present study was to evaluate how the molecules present in the Salvia miltiorrhiza extract could interfere with the binding between SARS-CoV-2 and the cellular ACE2 receptors, and to determine how this extract could interfere with the inflammatory response of blood mononuclear cells (PBMCs). These in vitro data demonstrate a potential of S. miltiorrhiza against COVID-19, consisting first of the blockade the binding of SARS-CoV-2 to ACE2 receptor and the mitigation of the inflammatory response from leukocytes by interfering with NFkB signaling.
The manuscript has scientific relevance and originality to be published.
The methodology and results are sufficiently well described and consistent with the discussion and conclusions.
My recommendation is for acceptance for publication.
Author Response
Reviewer #1
The purpose of the present study was to evaluate how the molecules present in the Salvia miltiorrhiza extract could interfere with the binding between SARS-CoV-2 and the cellular ACE2 receptors, and to determine how this extract could interfere with the inflammatory response of blood mononuclear cells (PBMCs). These in vitro data demonstrate a potential of S. miltiorrhiza against COVID-19, consisting first of the blockade the binding of SARS-CoV-2 to ACE2 receptor and the mitigation of the inflammatory response from leukocytes by interfering with NFkB signaling.
Comment 1. The manuscript has scientific relevance and originality to be published.
The methodology and results are sufficiently well described and consistent with the discussion and conclusions.
My recommendation is for acceptance for publication.
Response 1. The authors would like to thank the Reviewer for his/her positive feed-back about the manuscript.
Reviewer 2 Report
After reviewing the manuscript entitled “Salvia miltiorrhiza Bunge as a Potential Natural Compound against COVID-19”, some modifications must be considered.
The topic of the manuscript is well chosen and argued, but some errors must be resolved before the manuscript is considered suitable for publication.
I would recommend writing the article in third person. Therefore, make changes to eliminate “we”, “our” or “us” … review the entire manuscript and keep consistency.
Abstract: Please write in italics (and with the author at least the first time mentioned) the scientific name of the plant. The same for the rest of the manuscript. For the rest of the manuscript this could be abbreviated (S. miltiorrhiza).
“In vitro” is also written in italics.
lines 32 and 33. Check text formatting
How has the dry extract been prepared? What kind of extract is it? extraction solvent? extraction yield? ... It should be explained how the extract was obtained.
Subsequently (Analysis of the Salvia miltiorrhiza Bunge Root Extract (UHPLC) section), it is mentioned the extraction of the extract with methanol. Why doesn't the extract dissolve directly?
Line 135: “root powder” is not in italics.
Line 194: Check text formatting
Does the control have no deviations? if different controls are not carried out, it is easy to obtain significant differences later … Having a zero deviation in the control makes statistical study very beneficial. Several controls must be done in the same experiment to observe deviations.
To improve it, a deeper discussion must take place; I would also recommend to include in the discussion section the activities already seen in others papers by the isolated phytochemicals present in the extract.
Finally, I recommend including a section with the conclusions of the study.
Author Response
Reviewer #2
Comment 1. After reviewing the manuscript entitled “Salvia miltiorrhiza Bunge as a Potential Natural Compound against COVID-19”, some modifications must be considered.
The topic of the manuscript is well chosen and argued, but some errors must be resolved before the manuscript is considered suitable for publication.
Reply 1. We thank the Reviewer for his/her constructive comments.
C2. I would recommend writing the article in third person. Therefore, make changes to eliminate “we”, “our” or “us” … review the entire manuscript and keep consistency.
R2. All the “we”, “our” or “us” … were removed from the manuscript and replaced by the passive form.
C3. Abstract: Please write in italics (and with the author at least the first time mentioned) the scientific name of the plant. The same for the rest of the manuscript. For the rest of the manuscript this could be abbreviated (S. miltiorrhiza).“In vitro” is also written in italics.
lines 32 and 33. Check text formatting
R3. Those changes have been done in the revised manuscript.
C4. How has the dry extract been prepared? What kind of extract is it? extraction solvent? extraction yield? ... It should be explained how the extract was obtained.
Subsequently (Analysis of the Salvia miltiorrhiza Bunge Root Extract (UHPLC) section), it is mentioned the extraction of the extract with methanol. Why doesn't the extract dissolve directly?
R4. We thank the Reviewer for this comment. The preparation of the extract has been detailed in the Methods section of the revised manuscript.
The dry extract was indeed “extracted” in methanol prior to UHPLC analysis to dissolve it in the same solution as the standards and the usual samples. We applied the internal extraction procedure of our usual samples (which are S.miltiorrhiza root powder) while the dry extract was rapidly dissolved.
C5. Line 135: “root powder” is not in italics.
Line 194: Check text formatting
R5. This is now corrected in the revised text.
C6. Does the control have no deviations? if different controls are not carried out, it is easy to obtain significant differences later … Having a zero deviation in the control makes statistical study very beneficial. Several controls must be done in the same experiment to observe deviations.
R6. In figures 3 and 5, results are expressed as absolute values (thus with values for bothe control and test conditions). In figures 1, 2 and 4, the results are indeed expressed as fold or % changes from the control in order to facilitate the visualization of the data. In those cases, the dispersion of the data are seen in the test conditions (and not in the control) conditions, but did not affect the statistics. Therefore, the figures were kept with the same format of results in the revised manuscript.
C7. To improve it, a deeper discussion must take place; I would also recommend to include in the discussion section the activities already seen in others papers by the isolated phytochemicals present in the extract.
Finally, I recommend including a section with the conclusions of the study.
R7. Following the Reviewer’s suggestion, the Discussion has been improved by integrating previously documented activities of the extract’ components as well as by adding a final conclusion.
Round 2
Reviewer 2 Report
The article has been significantly improved with the revision changes.
It is possible that the analysis does not change, but by making different controls you can also express the percentage of change between them. This should be taken into account.